# Contraction reserve in high resolution manometry is correlated with lower esophageal acid exposure time in patients with normal esophageal motility: A retrospective observational study

**Yaoyao Lu[ID], Linling Lv, Jinlin Yang, Zhihui Yi[ID]***

Department of Gastroenterology and Hepatology, West China Hospital, Sichuan University, Chengdu, China

* yizhihuiHX@163.com

**Data Availability Statement:** All relevant data are within the paper and its Supporting Information files.

## Abstract

### Background

In high resolution manometry (HRM), distal contractile integral post multiple rapid swallow augmentation is considered as contraction reserve. The relationship between contraction reserve and esophageal acid reflux remains unclear. The aim of this study was to explore the correlation between contraction reserve and esophageal acid exposure in ineffective esophageal motility (IEM) and normal HRM.

### Methods

Patients who underwent HRM and ambulatory reflux monitoring were retrospectively screened. Those with diagnosis of normal HRM or IEM were included in the analysis. The proportion of patients with abnormal acid exposure time (AET) was compared between patients with and without contraction reserve. Multivariate regression analysis was performed to determine the predictors of abnormal AET and contraction reserve.

### Results

A total of 338 patients, including 264 normal HRM and 74 IEM, were included in the analysis. In patients with normal HRM, proportion of abnormal total AET (AET > 6.0%) was significantly lower in patients with supine contraction reserve than patients without contraction reserve (13.85% vs. 24.63%, $p = 0.027$). Multivariate regression analysis showed that supine contraction reserve could independently predict abnormal total AET (OR = 0.468, 95% CI: 0.249–0.948, $p = 0.034$), while upright contraction reserve trended strongly (OR = 0.558, 95% CI: 0.290–1.071, $p = 0.079$). Subgroup analysis showed that upright contraction reserve was an independent predictor of abnormal total AET in patients with 50–70% infective swallows (OR = 0.205, 95% CI: 0.051–0.821, $p = 0.025$), whereas supine contraction reserve did not have predictive value ($p = 0.359$).

**Funding:** The author(s) received no specific funding for this work.

**Competing interests:** The authors have declared that no competing interests exist.

## Conclusions

Supine contraction reserve correlates with esophageal acid reflux in patients with normal HRM, while only upright contraction reserve correlates with esophageal acid reflux in patients with infective swallows of 50–70%.

## Introduction

Ineffective esophageal motility (IEM) is the most common esophageal dysmotility diagnosed by high-resolution manometry (HRM) [1]. It is characterized by hypomotility of the esophagus with normal lower esophageal sphincter (LES) relaxation. According to Chicago classification version 4.0 (CCv4.0), IEM is defined as more than 70% ineffective swallows or at least 50% failed swallows with normal integrated relaxation pressure (IRP) on HRM [2]. The clinical implications of IEM remain obscure. It has been reported to be associated with impaired esophageal clearance and implicated in the pathophysiology of gastroesophageal reflux disease (GERD) [3]. Compared with patients with normal esophageal motility, those with IEM have a higher proportion of pathological acid reflux [4, 5]. However, IEM has not been consistently related to GERD. It has also been observed in asymptomatic healthy populations and patients with physiological acid reflux [6, 7]. Hence, in CCv4.0, patients with ineffective swallows of 50–70% are considered inconclusive for a diagnosis as IEM to further sift out clinically relevant IEM. Multiple rapid swallows (MRS), a provocative test that entails five swallows of 2 mL liquid at 2–3 s intervals, might be associated with esophageal acid reflux and provide supportive evidence for a diagnosis of IEM in these patients [2].

An intact response to MRS is defined as the absence of esophageal body contractility (distal contractile integral (DCI) <100 mmH-g•s•cm) with complete deglutitive inhibition of the LES during the repetitive swallows and presence of contraction reserve (DCI post-MRS greater than single swallow mean DCI) [2]. Previous studies indicated that the absence of contraction reserve might contribute to the pathophysiology of GERD. However, the results are conflicting and limited. One study reported that the absence of supine contraction reserve was an independent predictor of pathological esophageal acid reflux in patients with infective swallows of 50–70% [8]. However, in another study, the author did not find a correlation between supine contraction reserve and esophageal acid reflux [9]. Upright contraction reserve in IEM has not yet been described. Swallowing in the upright position is thought to more closely simulate real-life behaviors and may be better tolerated. Therefore, in CCv4.0, study in upright position was included in the standard HRM protocol [2]. The correlation between upright contraction reserve and esophageal acid reflux remains unclear.

In this study, we included more patients and retrospectively investigated the correlation between contraction reserve in both the supine and upright positions and esophageal acid reflux in patients with IEM and normal esophageal motility.

## Methods

### Ethics statement

This is a retrospective observational study. The study protocol was approved by Ethics Committee of West China Hospital of Sichuan University (approval No.2022-1644) and was granted a waiver of informed consent from all enrolled participants because of the retrospective study design. The study was conducted according to the Declaration of Helsinki. All data

were identified and analyzed anonymously. The study was conducted between November 2022 and March 2023 and study relevant data were collected between November 2022 and January 2023.

## Patients

Consecutive adult patients who underwent both HRM and ambulatory reflux monitoring between July 2021 and July 2022 at West China Hospital with a HRM diagnosis of normal HRM or IEM according to the CCv.4.0 criteria were included. Exclusion criteria were as follows: (i) esophageal and/or cardia tumor; (ii) grade C or D esophagitis according to the Los Angeles classification system; (iii) absent contractility; and (iv) missing MRS data. Baseline characteristics including age, sex, and body mass index (BMI) were collected and compared between groups.

## High-resolution manometry

HRM was performed following a protocol recommended by CCv4.0. This includes a 20–30 s quiet landmark phase, ten 5 mL liquid single swallows, and at least one MRS in the supine position following a 20–30 s quiet landmark phase, five 5 mL liquid single swallows, and at least one MRS in the upright position. All HRMs were performed after an overnight fast using a 36-sensor solid state HRM catheter (Manosan, Medronic Inc., Heerlen, Netherlands). MRS was performed as five swallows of 2 mL water with intervals of 2–3 s between swallows and repeated up to three times if there was a failed attempt or an abnormal contractile response in most patients. The mean basal esophagogastric junction (EGJ) pressure, DCI of each wet swallow, and DCI post-MRS (MRS-DCI) were calculated. The esophageal motor pattern was categorized as normal HRM and IEM diagnosed based on the supine position. A diagnosis of IEM required at least 50% failed swallows (DCI <100 mmHg·cm·s) or >70% ineffective swallows (including failed, weak, or fragmented swallows). Patients with ineffective swallows of not more than 70% and failed swallows of less than 50% are considered as normal HRM. The presence of contraction reserve was defined as MRS-DCI greater than the single swallow mean DCI. EGJ morphology was categorized based on the spatial relationship between LES and crural diaphragm (CD): Type I EGJ morphology is defined as complete overlap of LES and CD; type II EGJ morphology with LES and CD separation 1–2 cm; type III EGJ morphology the separation >2 cm [10].

## Esophageal 24-hour pH monitoring

Acid suppressants were ceased at least 7 days prior to esophageal 24-hour pH monitoring. The pH monitoring catheter (VersaFlex, Medronic Inc., Heerlen, Netherlands) was placed in the esophagus for at least 22 h. The pH sensor was placed 5 cm proximal to the LES, which was located using HRM. Total AET >6% was considered pathological total AET [11]. Supine AET >2.0% was considered pathological supine AET [8].

## Statistical analysis

Continuous data were described as the mean ± standard deviation or median with interquartile range depending on their distribution. Categorical variables were reported as counts with proportions. Continuous variables conforming to normal distribution assumptions were compared between groups using 2-tailed Student's t test. Continuous variables that did not conform to normal distribution assumptions were compared between groups using the Mann–Whitney U test. Categorical data were compared between groups using a chi-squared test.

Multivariate logistic regression using the Enter method was performed to identify potential predictors for abnormal AET with the following variables: EGJ morphology, mean basal EGJ pressure and contraction reserve. Multivariate logistic regression was also performed to identify potential predictors for contractile reserve using the forward method including the following variables: age, sex, total AET classification (total AET $\geq$ 6.0% = 1, total AET <6.0% = 0) and BMI. Statistical significance was set at $p<$ 0.05. All statistical analyses were performed using SPSS 26.0 software.

## Results

### Demographic data and patient diagnoses

A total of 350 patients with diagnosis of normal HRM or IEM were screened, and 12 of them were excluded for esophageal cancer (n = 2), grade C esophagitis according to the Los Angeles classification system (n = 1) and missing MRS data (n = 9). Finally, 338 patients fulfilled the inclusion criteria, including 264 (78.11%) normal HRM and 74 (21.89%) IEM cases (Fig 1).

The baseline characteristics of the patients are summarized in Table 1. Compared with patients with normal HRM, patients with IEM had a significantly higher proportion of males ($p$ = 0.044) and lower mean basal EGJ pressure ($p$ = 0.006). A significantly higher proportion of abnormal total AET ($p$ = 0.016) and supine AET ($p$ = 0.013) were observed in patients with IEM compared with patients with normal esophageal motility.

### Contraction reserve and acid burden

In the supine position, the proportions of patients with contraction reserve were 49.24% (130/264) and 35.14% (26/74) in the normal HRM group and IEM group respectively. In patients without contraction reserve, 96.2% (175/182) had 3 MRS.

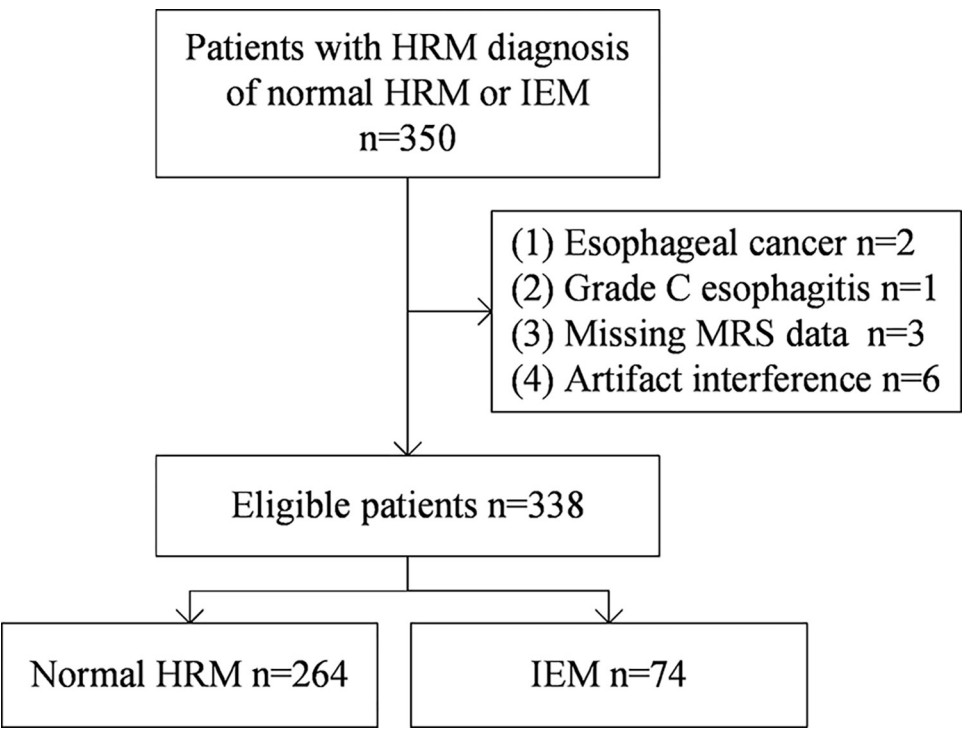

**Fig 1. Flow chart of patient selection.**

**Table 1. Baseline characteristics of the study population.**

| | Normal (n = 264) | IEM (n = 74) | p value |
|---|---|---|---|
| **Age (y), mean±SD** | 47.18 ± 11.48 | 45.49 ± 11.68 | 0.264 |
| **Gender (Male), n(%)** | 108(40.91%) | 40(54.05%) | 0.044 |
| BMI (Kg/m$^2$), mean±SD | 22.42 ± 3.14 (n = 262) | 21.76 ± 2.66 (n = 69) | 0.108 |
| **Mean EGJ basal pressure (mmHg), mean±SD** | 21.65 ± 10.98 | 17.16 ± 9.28 | 0.01 |
| **EGJ morphology** | | | 0.771 |
| **Type I EGJ, n(%)** | 210(79.55%) | 60(81.08%) | |
| **Type II/III EGJ, n(%)** | 54(20.45%) | 14(18.92%) | |
| **Abnormal AET** | | | |
| **Total AET>6.0%, n(%)** | 51(19.32%) | 24(32.43%) | 0.016 |
| **Supine AET>2.0%, n(%)** | 36(13.25%) | 19(25.68%) | 0.013 |

In the upright position, the proportions of upright contraction reserve were 55.68% (147/264) and 32.43% (24/74) in the normal HRM group and IEM group respectively. In patients without contraction reserve, 77.2% (129/167) had 3 MRS. The concordance of contraction reserve presence in the upright position and supine position was 64.79%. Cohen's kappa was 0.298 (p <0.001).

In the normal HRM group, patients with supine contraction reserve showed a lower proportion of abnormal total AET (13.85% vs. 24.63%, $p$ = 0.027), but a similar proportion of abnormal supine AET (10.77% vs. 16.42%, $p$ = 0.181) compared with those without contraction reserve (Fig 2A). In patients with and without upright contraction reserve, there were not significant difference in proportion of total AET (15.65% vs. 23.93%, $p$ = 0.09) and supine AET (11,56% vs. 16.24%, $p$ = 0.272, Fig 2B).

In the IEM group, there were no significant differences in proportion of both abnormal total AET (30.77% vs. 33.33%, $p$ = 0.822) and abnormal supine AET (26.92% vs. 25.00%, $p$ = 0.857) between patients with and without supine contraction reserve. Similarly, the proportions of abnormal total AET (29.17% vs. 34.0%, $p$ = 0.678) and supine AET (25% vs. 26.0%, $p$ = 0.509) were also similar between patients with and without uptight contraction reserve (Fig 2B). Both MRS-DCI and the ratio of MRS-DCI to mean single swallows DCI between

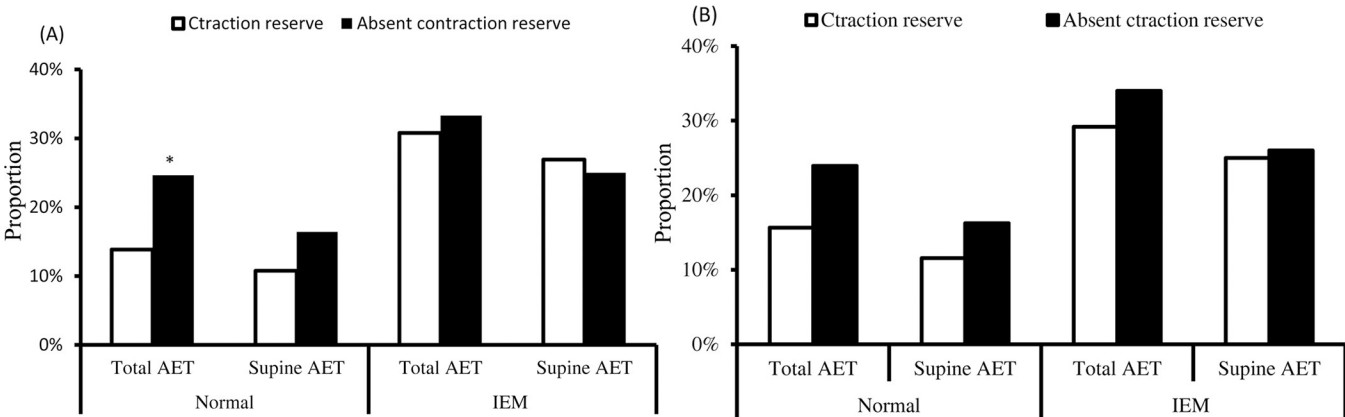

**Fig 2. Comparison of abnormal acid exposure time (AET) between patients with and without contraction reserve in each group.** (A) Patients with supine contraction reserve have lower proportion of abnormal total AET compared with those without contraction reserve in patients with normal esophageal motility, but not with IEM. (B) Proportions of abnormal AET in patients with and without upright contraction reserve were similar between groups. *$p$<0.05 compared to patients with contraction reserve within each category. IEM: ineffective esophageal motility.

patients with normal AET and abnormal AET were not significantly different in the IEM group ($P>0.05$, S1 File).

## Multivariate logistic regression analysis of AET

In patients with normal HRM, supine contraction reserve independently predicted abnormal total AET ($p = 0.034$, Table 2). Type II/III EGJ morphology was an independent predictor of both abnormal total AET ($p = 0.001$) and abnormal supine AET ($p = 0.006$, Table 2).

To evaluate the impact of contraction reserve on abnormal esophageal acid reflux in patients with 50–70% ineffective swallows, we further performed a subset analysis. After controlling for EGJ morphology, upright contraction reserve was an independent predictor of abnormal total AET (OR = 0.205, 95% CI: 0.051–0.821, $p = 0.025$), whereas supine contraction reserve lacked predictive value ($p = 0.359$, S2 File).

In the IEM group, since the chi-squared test showed that contraction reserve was not associated with AET, contraction reserve was not included in the multivariate regression analysis. Results showed that mean basal pressure was an independent predictor of abnormal total AET (OR = 0.914, 95% CI: 0.845–0.988, p = 0.024), while EGJ morphology could not predict either abnormal total AET or abnormal supine AET (S2 File).

## Logistic regression analysis of contraction reserve

Multivariate logistic regression showed that age was an independent predictor of both supine contraction reserve (OR = 0.972, 95% CI: 0.952–0.991, $p = 0.005$) and upright contraction reserve (OR = 0.963, 95% CI: 0.994–0.982, $p<0.001$), while sex was an independent predictor of presence of supine contraction reserve (OR = 2.153, 95% CI: 1,346–3.444, $p = 0.001$) after controlling for AET. BMI could not predict contraction reserve ($p >0.05$, S2 File).

## Discussion

In this retrospective study, we evaluated the correlation between contraction reserve and esophageal acid exposure. We found that supine contraction reserve is associated with lower esophageal acid exposure in normal esophageal motility patients, but not in IEM patients. Only upright contraction reserve, but not supine, is associated with lower esophageal acid exposure in patients with 50–70% ineffective swallows. Male and young patients are more likely to have contraction reserve than female and old patients.

Several studies have explored the clinical significance of contraction reserve. It has been reported that supine contraction reserve exists in 70–80% of healthy volunteers but exists in only 46–50% of patients with normal esophageal motility and 48% of patients with IEM [12, 13]. Hence, it is supposed that the absence of contraction reserve may indicate potential

**Table 2. Multivariate logistic regression to identify predictors of AET in normal HRM patients.**

|  | Odds ratio | 95% CI | *p* value |
|---|---|---|---|
| Total AET |  |  |  |
| Mean EGJ basal pressure | 0.967 | 0.934–1.001 | 0.058 |
| EGJ morphology | 3.568 | 1.706–7.416 | 0.001 |
| Supine contraction reserve | 0.486 | 00.249–0.948 | 0.034 |
| Supine AET |  |  |  |
| Mean EGJ basal pressure | 0.976 | 0.939–1.014 | 0.205 |
| EGJ morphology | 3.130 | 1.377–7.112 | 0.006 |
| Supine contraction reserve | 0.632 | 0.301–1.326 | 0.225 |

abnormality of esophageal motility. In the present study, we found that patients without contraction reserve have higher proportion of abnormal AET, which also confirms this hypothesis. Several studies reported the correlation between contraction reserve and esophageal acid exposure previously. It is reported that in patients with normal esophageal motility, the ratio of MRS-DCI to mean single swallows DCI was higher in patients with physiological acid reflux than in patients with pathological acid reflux [14]. In another study, Farhan et al. reported that the proportion of pathological acid reflux was significantly lower when supine contraction reserve was present compared with when contraction reserve was absent in patients with 50–70% ineffective swallow, but not in patients with IEM [8]. In contrast to previous findings, however, we did not find an association between supine contraction reserve and abnormal AET in patients with 50–70% ineffective swallows. This difference may be explained by two reasons. First, it may be due to the difference in MRS number. It has been reported that the reproducibility of the contraction reserve was poor when only one MRS was conducted (kappa = 0.15) [15]. In CCv4.0, it is recommended that MRS can be repeated up to three times if there is a failed attempt or an abnormal contractile response [2]. In the two studies mentioned above, only one MRS was performed. While in the present study, 96.2% of patients without supine contraction reserve underwent 3 MRSs to confirm the results. Another possible explanation for the inconsistent results may be the different characteristics of the study populations.

We further explored the correlation between upright contraction reserve and esophageal acid reflux, which has not yet been described. Study in upright position has been included in the standard HRM protocol as recommended by CCv4.0 [2]. In the upright position, the presence of gravity facilitates esophageal emptying and results in reduced DCI and LES pressure [16], and cardiovascular artifacts may also be reduced [17]. Discrepancies in HRM diagnosis between upright and supine position are not rare [16]. Final HRM diagnosis is recommended to be made based on both supine and upright position which can decrease clinically irrelevant diagnosis [2]. Similarly, diagnostic agreement of contraction reserve in the upright and supine position is moderate, only 64.5% in the present study. Interestingly, we found that upright contraction reserve could predict abnormal acid reflux in patients with 50–70% ineffective swallows while supine contraction could not.

Contraction reserve is recommended as supportive evidence for a diagnosis of IEM in patients with 50–70% ineffective swallows [2]. However, we think it should be cautious. First, besides contraction reserve, several parameters related to IEM can even better predict abnormal esophageal acid reflux, such as the postreflux swallow induced peristaltic wave index and mean LES basal pressure [18, 19]. However, none of them are considered as supportive evidence for the diagnosis of IEM. Second, even 20–30% of healthy volunteers lack contraction reserve [12, 13]. Finally, we believe that the normal value of the ratio between MRS-DCI and mean single swallows DCI varies among populations with different demographic characteristics. It has been reported that normative thresholds of metrics in HRM vary according to age, obesity, ethnicity and body position [20]. In the present study, we also found that age and sex were correlated with contraction reserve. Younger age and male patients were more likely to have contraction reserve than older and female patients. This indicates that the predictive ability of contraction reserve for abnormal AET may vary among populations with different characteristics. The clinical implication of contraction reserve in patients with IEM remains obscure.

The present study has several limitations. First, as mentioned above, the optimal number of MRS to define contraction reserve is controversial, especially in the upright position. MRS in the upright position is not included in the standard HRM protocol recommended by CCv4.0. In the present study, nearly 80% of patients without uptight contraction reserve had 3 MRS.

Second, we did not include healthy volunteers to determine the normative threshold for ratio between MRS-DCI and mean single swallows DCI. Finally, we did not explore the relationship between contraction reserve and esophageal bolus clearance. We were also not able to compare treatment response in patients with and without contraction reserve.

## Conclusions

We demonstrated that both supine and upright contraction reserve correlate with esophageal acid reflux in patients with normal esophageal motility but not IEM. Only upright contraction reserve correlates with esophageal acid reflux in patients with 50–70% ineffective swallows. In addition to AET, age and sex are also related to contraction reserve. The definition of contraction reserve might be redefined according to age and sex. Further studies on the prognosis and treatment response of the absence of contraction reserve are also needed.

## Supporting information

**S1 File. Comparison of distal contractile integral between patients with and without pathological acid reflux.**
(XLSX)

**S2 File. Full results of logistic regression analysis.**
(XLSX)

## Acknowledgments

We thank Yu Zeng for his efforts in our department.

## Author Contributions

**Conceptualization:** Yaoyao Lu, Jinlin Yang, Zhihui Yi.

**Data curation:** Yaoyao Lu, Linling Lv.

**Formal analysis:** Yaoyao Lu.

**Methodology:** Yaoyao Lu.

**Supervision:** Zhihui Yi.

**Validation:** Yaoyao Lu.

**Visualization:** Yaoyao Lu, Zhihui Yi.

**Writing – original draft:** Yaoyao Lu, Jinlin Yang, Zhihui Yi.

**Writing – review & editing:** Yaoyao Lu, Jinlin Yang, Zhihui Yi.

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
