## [Decision Letter · Decision Letter 0]

29 Jun 2023

PONE-D-23-05480Contraction reserve in high resolution manometry is correlated with lower esophageal acid exposure time in patients with normal esophageal motility： A restrospective observational studyPLOS ONE

Dear Dr. Yi,

Thank you for submitting your manuscript to PLOS ONE. After careful consideration, we feel that it has merit but does not fully meet PLOS ONE’s publication criteria as it currently stands. Therefore, we invite you to submit a revised version of the manuscript that addresses the points raised during the review process.

ACADEMIC EDITOR:

This is an important study giving interesting data about HRM examination of esophagus. Before considering for publication, the issues mentioned by reviewers should be carefully revised. Additionally, please verify the value of contraction reserve in normal HRM clearly, and give the clinical relevance of contraction reserve (how does it effect the management choice, etc).

We look forward to receiving your revised manuscript.

Kind regards,

Ozlem Boybeyi-Turer

Academic Editor

PLOS ONE

Journal Requirements:

Additional Editor Comments:

This is an important study giving interesting data about HRM examination of esophagus. Before considering for publication, the issues mentioned by reviewers should be carefully revised. Additionally, please verify the value of contraction reserve in normal HRM clearly, and give the clinical relevance of contraction reserve (how does it effect the management choice, etc).

Best Regards

Reviewers' comments:

Reviewer's Responses to Questions

**Comments to the Author**

1. Is the manuscript technically sound, and do the data support the conclusions?

Reviewer #1: Partly

Reviewer #2: Yes

2. Has the statistical analysis been performed appropriately and rigorously? 

Reviewer #1: Yes

Reviewer #2: I Don't Know

3. Have the authors made all data underlying the findings in their manuscript fully available?

Reviewer #1: Yes

Reviewer #2: No

4. Is the manuscript presented in an intelligible fashion and written in standard English?

Reviewer #1: No

Reviewer #2: Yes

5. Review Comments to the Author

Reviewer #1: Lu et al. present a series of esophageal high resolution manometry to correlate acid exposure time with contraction reserve in patients with normal versus ineffective motility patterns at the manometry.

The manuscript present interetsing data but it may be interpretate with caution. I personaly do not believe that contaction reserve is significant. It is one more of the temptatives to force parameters using HRM. I read the manuscript carefully trying to find meaning to the usefulness of this parameter but I could not find. Moreover, I was expecting the authors to discuss the topics I am presenting below.

1) What is the value of contraction reserve in normal HRM? Is the esophagus supposed to contract eve more than he normaly contracts? If I have normal motility is it not an assumption that I have reserve?

2) Why would reserve affect AET in normal peristalsis if esophageal clearace is theoretically normal?

3) Contraction reserve could be useful in patients with IEM to show that despite abnormal motility with standard stimulation, there is still working muscle with overload. The authors did not saw a correlation of this with GERD. Than waht is the use of this parameter?? The authors are deyifing this parameter.

The manuscript needs extensive rediscussion in order to make sense of the data.

Minor comments:

1) Languae needs some review in terms of scientific soundable. E.g. "Ineffective esophageal motility (IEM) is most often encountered in patients referred for high-resolution manometry"  that is obvious. If the diagnosis is made by HRM how can patients be diagnosed with IEM otherwise???????

2) Why esophagitis LA C was excluded?

Reviewer #2: This study show that both supine and upright esophageal contraction reserve correlates with abnormal esophageal acid exposure in patients with normal esophageal motility on HRM, while only upright contraction reserve correlates with acid esposure in patients with 50-70% ineffective swallows. These are important results that may contribute to improve patient selection for medical versus surgical therapy. The methodology of the study is impeccable and the manuscript is quite well written. I have the following minor comments: 1)In the discussion section, page 3,, lines 232 on, please briefly comment a recent review paper (Riva CG, et al, J Neurogastro Motility 2020;26(3):335-343) focusing on the drawbacks of conducting HRM studies in the supine position only. 2)Please check the following items: page 8, line 140, "...patients had...", line 147, delete "...tendency toward..."; page 9, line 162, delete "tendency toward", line 168, delete "...trended strongly"; page 13, line 239, "...patients had...; page 14, line 270, "...the present study does not support....

6. PLOS authors have the option to publish the peer review history of their article (what does this mean?). If published, this will include your full peer review and any attached files.

Reviewer #1: No

Reviewer #2: No

---

## [Author Response · Author response to Decision Letter 0]

11 Aug 2023

Academic Editor: 

This is an important study giving interesting data about HRM examination of esophagus. Before considering for publication, the issues mentioned by reviewers should be carefully revised. Additionally, please verify the value of contraction reserve in normal HRM clearly, and give the clinical relevance of contraction reserve (how does it effect the management choice, etc).

Response: In the revised manuscript, we discussed the value of contraction reserve in normal HRM and clinical relevance of contraction reserve, which can be found in page11, line 202-208 and page 13, line 236-237. 

Reviewer 1#

1) What is the value of contraction reserve in normal HRM? Is the esophagus supposed to contract even more than he normally contracts? If I have normal motility is it not an assumption that I have contraction reserve?

Response: It is reported that supine contraction reserve is existed in 70-80% of healthy volunteers while exist only in 46-50% of patients with normal HRM. Hence, it is supposed that the absence of contraction reserve may indicate potential damage to esophageal contractility. In the present study, we found that contraction reserve can independently predict abnormal AET. We discussed this question in the discussion part in the revised manuscript (page11, line 202-208). 

2) Why would reserve affect AET in normal peristalsis if esophageal clearance is theoretically normal?

Response: Impaired esophageal clearance is not the only reason for abnormal AET. Transient lower esophageal sphincter (LES) relaxation, lower mean basal LES pressure, decreased postreflux swallow-induced peristaltic wave index, etc., are all risks factors for pathological acid exposure. The mechanism by which contraction reserve affects AET is not clear at present. We suppose that in some patients, the absent of contraction reserve may indicate potential abnormality of esophageal motility. However, in other patients, it may not have specific significance because 20-30% of healthy volunteers do not have contraction reserve. 

3) Contraction reserve could be useful in patients with IEM to show that despite abnormal motility with standard stimulation, there is still working muscle with overload. The authors did not saw a correlation of this with GERD. Then what is the use of this parameter?? The authors are deyifing this parameter.

Response: We designed this study because we doubted the value of contraction reserve in predicting abnormal acid reflux. We finally found a correlation between contraction reserve and abnormal AET in patients with normal esophageal motility, but not IEM. The mechanism by which contraction reserve affects AET is not clear. 

4) Language needs some review in terms of scientific soundable. E.g. "Ineffective esophageal motility (IEM) is most often encountered in patients referred for high-resolution manometry"  that is obvious. If the diagnosis is made by HRM how can patients be diagnosed with IEM otherwise?

Response: We have carefully revised the language of the entire manuscript. This sentence has been revised as “Ineffective esophageal motility (IEM) is the most common esophageal dysmotility diagnosed by high-resolution manometry.”

5) Why esophagitis LA C was excluded?

Response: At the design stage, we excluded LA C esophagitis to avoid the possibility that severe esophagitis induced esophageal dysmotility, which might influence the study results. The only LA C esophagitis is normal HRM with supine contraction reserve absent and uptight contraction reserve present. The total AET and supine AET were 1.1% and 1.2% respectively. When this case is included in the analysis, the result is not changed. Therefore, we did not revise the study protocol to include this case. 

Reviewer 2#

1) In the discussion section, page 3, lines 232 on, please briefly comment a recent review paper (Riva CG, et al, J Neurogastro Motility 2020;26(3):335-343) focusing on the drawbacks of conducting HRM studies in the supine position only. 

Response: We have revised the manuscript accordingly. Discrepancies in HRM diagnosis between upright and supine position are not rare. Final HRM diagnosis is recommended to be made based on both supine and upright position which can decrease clinically irrelevant diagnosis. The revised text can be found in page 12, line 229-232.

2) Please check the following items: page 8, line 140, "...patients had...", line 147, delete "...tendency toward..."; page 9, line 162, delete "tendency toward", line 168, delete "...trended strongly"; page 13, line 239, "...patients had...; page 14, line 270, "...the present study does not support....

Response: We have revised the manuscript accordingly.

---

## [Decision Letter · Decision Letter 1]

21 Aug 2023

Contraction reserve in high resolution manometry is correlated with lower esophageal acid exposure time in patients with normal esophageal motility： A restrospective observational study

PONE-D-23-05480R1

Dear Dr. Yi,

We’re pleased to inform you that your manuscript has been judged scientifically suitable for publication and will be formally accepted for publication once it meets all outstanding technical requirements.

Kind regards,

Ozlem Boybeyi-Turer

Academic Editor

PLOS ONE

Additional Editor Comments (optional):

Reviewers' comments:

Reviewer's Responses to Questions

**Comments to the Author**

1. If the authors have adequately addressed your comments raised in a previous round of review and you feel that this manuscript is now acceptable for publication, you may indicate that here to bypass the “Comments to the Author” section, enter your conflict of interest statement in the “Confidential to Editor” section, and submit your "Accept" recommendation.

Reviewer #1: All comments have been addressed

Reviewer #2: All comments have been addressed

2. Is the manuscript technically sound, and do the data support the conclusions?

Reviewer #1: Yes

Reviewer #2: Yes

3. Has the statistical analysis been performed appropriately and rigorously? 

Reviewer #1: Yes

Reviewer #2: Yes

4. Have the authors made all data underlying the findings in their manuscript fully available?

Reviewer #1: Yes

Reviewer #2: (No Response)

5. Is the manuscript presented in an intelligible fashion and written in standard English?

Reviewer #1: Yes

Reviewer #2: Yes

6. Review Comments to the Author

Reviewer #1: The authors addressed well all comments.

No further comments.

Reviewer #2: (No Response)

7. PLOS authors have the option to publish the peer review history of their article (what does this mean?). If published, this will include your full peer review and any attached files.

Reviewer #1: No

Reviewer #2: **Yes: **Luigi Bonavina

---

## [Editor Report · Acceptance letter]

26 Oct 2023

PONE-D-23-05480R1 

Contraction reserve in high resolution manometry is correlated with lower esophageal acid exposure time in patients with normal esophageal motility: A retrospective observational study 

Dear Dr. Yi:

I'm pleased to inform you that your manuscript has been deemed suitable for publication in PLOS ONE. Congratulations! Your manuscript is now with our production department. 

Kind regards, 

on behalf of

Professor Ozlem Boybeyi-Turer 

Academic Editor

PLOS ONE